# Influence of Te-Incorporated LaCoO_3_ on Structural, Morphology and Magnetic Properties for Multifunctional Device Applications

**DOI:** 10.3390/ijms241210107

**Published:** 2023-06-14

**Authors:** Jhelai Sahadevan, P. Sivaprakash, S. Esakki Muthu, Ikhyun Kim, N. Padmanathan, V. Eswaramoorthi

**Affiliations:** 1Department of Physics, Karpagam Academy of Higher Education, Coimbatore 641021, India; jhelaidev@gmail.com (J.S.); padmanmsc@gmail.com (N.P.); 2Department of Mechanical Engineering, Keimyung University, Daegu 42601, Republic of Korea; siva.siva820@gmail.com; 3Micro-Nano System Centre, Tyndall National Institute, University College Cork, T12R5CP Cork, Ireland; 4Department of Physics, Karpagam College of Engineering, Coimbatore 641032, India; physicseswar@gmail.com

**Keywords:** rhombohedral, hydrothermal synthesis, chemical reduction, oxygen deficiency, low spin state, ferromagnetism

## Abstract

A high perovskite activity is sought for use in magnetic applications. In this paper, we present the simple synthesis of (2.5% and 5%) Tellurium-impregnated-LaCoO_3_ (Te-LCO), Te and LaCoO_3_ (LCO) by using a ball mill, chemical reduction, and hydrothermal synthesis, respectively. We also explored the structure stability along with the magnetic properties of Te-LCO. Te has a rhombohedral crystal structure, whereas Te-LCO has a hexagonal crystal system. The reconstructed Te was imbued with LCO that was produced by hydrothermal synthesis; as the concentration of the imbuing agent grew, the material became magnetically preferred. According to the X-ray photoelectron spectra, the oxidation state of the cobaltite is one that is magnetically advantageous. As a result of the fact that the creation of oxygen-deficient perovskites has been shown to influence the mixed (Te^4+/2−^) valence state of the incorporated samples, it is abundantly obvious that this process is of utmost significance. The TEM image confirms the inclusion of Te in LCO. The samples start out in a paramagnetic state (LCO), but when Te is added to the mixture, the magnetic state shifts to a weak ferromagnetic one. It is at this point that hysteresis occurs due to the presence of Te. Despite being doped with Mn in our prior study, rhombohedral LCO retains its paramagnetic characteristic at room temperature (RT). As a result, the purpose of this study was to determine the impacts of RT field dependency of magnetization (M-H) for Te-impregnated LCO in order to improve the magnetic properties of RT because it is a low-cost material for advanced multi-functional and energy applications.

## 1. Introduction

Perovskites are one of the most fascinating types of solid materials, exhibiting a wide range of physical events and characteristics. Extensive research has been conducted on ABO_3_-type perovskites with the general formula Ln_1−x_A_x_MO_3_ or LnB_x_M_1−x_O_3_ (Ln-Lanthanides and M-dopant). For decades, researchers have been intrigued by nano crystallite magnetic cobaltite due to its outstanding magnetic and electric capabilities. Single-phase LaCoO_3_ perovskite is a good example of ceramic materials and is utilised in essential applications such as ZrB_2_ + 10 wt.% SiC for leading edges and nose cones in hypersonic vehicles and LaCoO_3_ for solid oxide fuel cell cathodes [1]. A wide variety of practical applications are made possible by many intrinsic perovskite materials features due to the continuous interaction between structure and properties. Ferroelectricity [2,3,4], semi-conductivity [5,6], superconductivity [5,6], piezoelectricity [7,8], thermoelectricity [9], colossal magnetoresistance, ferromagnetism [10], and half-metallic transport [11,12] are just some of the fascinating physical and chemical properties of perovskites. These oxides are increasingly being used in electronic and magnetic materials, automotive exhaust, water splitting catalysts, fuel cells, battery electrode materials [13], gas sensors, humidity sensors, microwave devices, high-density data storage, magnetic ferrofluids, magnetic switches, MRI, high-frequency, and power devices are among the applications for these materials [2]. As demonstrated by the discovery of superconductivity in Na_0.3_CoO_2_.1.3H_2_O, the fact that cobalt cations can assume multiple oxidation and spin states is the root cause of the wide range of observable physical features of cobaltites [14]. Specifically, the LCO perovskite is a classic example of thermally aided spin state transitions of trivalent cobalt [15,16]. Integration of divalent/trivalent or magnetic/nonmagnetic dopant ions into a lattice results in a drastic alteration of the structure and other properties defined by cation distribution. A dopant choice is needed to get the desired improvement in the unaltered perovskite cobaltite. In particular, the shift from paramagnetic (PM) to ferromagnetic (FM) at the Curie temperature (T_C_) and the accompanying insulator–metal transition (T_IM_) in the case of manganite and cobaltite have been known for some time. These include high conductivity, magnetic properties, excellent performance as a cathode or an anode, a high Seebeck coefficient, and various forms of oxygen vacancy ordering. The LaCoO_3_ perovskite, for instance, is a famous example of thermally aided trivalent cobalt spin state transitions. It has a nonmagnetic insulator ground state with only low-spin (LS) Co^3+^, but its magnetic susceptibility rises with temperature up to 100 K due to a transition from the LS to the intermediate-spin (IS) state. A second spin-state transition is then inferred from a shift in transport characteristics above 500 K from an activated regime (0.1 eV at 100 K) to a metallic regime (~1 mΩ cm), which corresponds to a conversion to high-spin (HS). The energetic proximity of the various Co^3+^ spin-states (LS, IS, and HS), as demonstrated by these two spin-state transitions, is also highlighted by the dramatic effect of a small amount of doping on physical attributes [17].

Ferroelastic materials having rhombohedral lattices are particularly fascinating. Stretching along one of the perovskite unit cell’s four body diagonals distorts the parent cubic structure, causing ferro elasticity [18]. As previously reported, LCO is ferromagnetic at low temperatures [19], and B site incorporation increases the magnetic properties. The complicated interplay between the interatomic exchange interaction energy (Δ_ex_) and the crystal field splitting energy (Δ_cf_) controls the active spin crossover of Co ion’s low and high spin states. A modest structural disruption brought on by strain can have a considerable effect since Δ_cf_ is particularly sensitive to changes in O-Co bond length and Co-O-Co bond angle [20,21].

Microwave heating, chemical co-precipitation, sol–gel auto-combustion [22], micro- and nano-emulsions, hydrothermal techniques, high-temperature breakdown, and reverse micelle have all been used to create cobaltite magnetic nanoparticles (NPs) [23,24,25]. The best way to produce LaCoO_3_ is through hydrothermal synthesis because it is easy, cheap, and safe for the environment. One of the most important aspects of our study was the hydrothermal method, which allowed us to produce superior perovskite precipitates with the required stoichiometry and microstructure. Hydrothermal synthesis of LCO by L. Tepech-Carrillo et al., followed by a study of LCO structure at different calcination temperatures, is elaborated. However, they did not look into LCO or Te-LCO composites’ structures or magnetic properties. Since Te (2.5% and 5%) in LaCoO_3_ modifies the magnetic properties, we are attracted to studying the magnetic properties and structural analysis of this material [26]. Unlike our prior work, in which we discovered no change in paramagnetic states at RT when doping Mn into LCO, we found that the Te inclusion in LCO has impacted the RT magnetic characteristics of LCO crystals. Therefore, we set out to improve LCO’s magnetic characteristics when it was exposed to RT. After discovering LT ferromagnetism in Mn-doped LCO, we set out to investigate whether or not the same phenomenon would occur in Te-incorporated LCO.

## 2. Results and Discussion

### 2.1. Powder X-ray Diffraction (P-XRD) of Te-Incorporated LaCoO_3_

For parent LCO and two different concentrations of 2.5% and 5%Te-LCO, the powder XRD patterns were taken. The standard single-phase crystallite perovskite structure of LCO (JCPDS 48-0123) can be used to assign all of the diffraction peaks of the prepared samples, proving that the perovskite structures are well preserved after Te incorporation, as shown in Figure 1. There were no peaks that could be attributed to impurities. Furthermore, an enlarged scale of the higher intensity diffraction peaks of the prepared samples (2θ range: 30–38°) is shown at the outset of Figure 1 and reveals a slight shift towards higher 2θ as the concentration of Te^2−^ (ionic radius 207 pm) increases, which has been attributed to Te’s larger ionic radius compared with that of Te. The crystal planes of rhombohedral LCO are marked in Figure 1 with reference to standard JCPDS Card No. 48-0123. To understand the reduction in TeO_2_, we took XRD for both parents and reduced Te, which is shown in Figure 2. TeO_2_ is successfully reduced to Te, as shown by the XRD data, which is supported by the ICSD collection codes 34-422 and 65-3048 for TeO_2_ and Te, respectively. The crystal structure transforms from tetragonal to hexagonal during the reduction of TeO_2_ to Te, as reported in the reference datasheet. Table 1 illustrates the lattice parameter, crystal system, space group, and crystallite size of TeO_2_, Te, LCO, 2.5% Te-LCO, and 5% Te-LCO. The average crystal size was determined using the Debye–Scherrer formula and the lattice parameters were determined using unit cell software [Name: Unit cell win, Version CCP14].

### 2.2. Raman Spectrum Analysis of Te-Incorporated LaCoO_3_

Hydrothermally produced LCO, chemically reduced Te, and Te-impregnated LCO are analysed for their chemical distribution using Raman spectroscopy. In Figure 3, Raman scattering measurements of LCO, Te, 2.5% Te-LCO, and 5% Te-LCO are displayed. The primeval LCO has weak Raman signals compared to Te, 2.5% Te-LCO, and 5% Te-LCO, whereas the 2.5% and 5% Te in cooperated LCO have good Te signals, which match the Te and have a slight shift and have a significant Raman characteristic peak. The Co-O-Co stretching mode in strongly deformed CoO_6_ may be related to the newly widened band at 516 cm^−1^ [27]. The E_g_ symmetry, which may be related to exterior mode (La-O) vibration, is responsible for the 116 and 140 cm^−1^ peaks [28,29]. Te exhibits strong Raman-active phonon modes as a result of its high atomic number and electronic polarizability. Three atoms make up each tellurium unit cell, which is arranged in an unending chain parallel to the c-axis. One A_1_ mode and two degenerate E modes, which are identified by rigid-chain rotation across the a- and b-axes and which fit into the Te lattice’s D_3_ symmetry group, are indicated by the Raman spectra. The week bands at 103 cm^−1^ are attributed to the E_(1)_ modes, which are responsible for the Raman spectra’s E_(1)_ bands. The E_(2)_ mode, which is primarily distinguished by asymmetries along the c-axis, corresponds to the bands at 141 cm^−1^. The broad bands are of the second order at 268.9 cm^−1^ [30,31,32]. Tellurium’s characteristic peaks at 122 and 141 cm^−1^ can be seen in both 2.5% Te-LCO and 5% Te-LCO, which corresponds to the A_1_ bond-stretching mode and two degenerate E bond-stretching modes, respectively [33]. There is a slight red shift for the high-intensity Te peak in Te-LCO, and increasing the Te concentrations broadens the peaks.

### 2.3. SEM and Energy-Dispersive X-ray (EDS) Analysis of Te-Incorporated LaCoO_3_

Te-LCO composites are created by impregnating tellurium into the pores of LCO, which successfully preserves the material’s irregular spherical shape. Figure 4a–e presents the surface morphology of TeO_2_, Te, LCO, 2.5% Te-LCO, and 5% Te-LCO, respectively. TeO_2_ and Te show (Figure 4a,b) that the reduction leads to the uniformly distributed plated nanostructure from mono-dispersed irregular microcrystals of about a few micrometers to nearly 500 nm. The hydrothermal synthesis of LCO (Figure 4c) leads to a well-dispersed spherical crystallite structure and the impregnating of Te in LCO (Figure 4d,e) has impacted slightly on the microstructure. The Te-impregnated LCO is well segregated, which is visible in the SEM micrograph. The EDAX spectrum of TeO_2_, Te, LCO, 2.5% Te-LCO, and 5% Te-LCO is revealed in Figure 4, which confirms the occurrence of constituent elements.

### 2.4. X-ray Photoelectron Spectroscopy (XPS) Studies of Te-Incorporated LaCoO_3_

X-ray photoelectron spectroscopy (XPS) is utilised to investigate the valence state of pure LCO and Te in-cooperated LCO with a Te content of 5 wt.%. The XPS survey spectra in Figure 5a,d demonstrate the presence of Co, O, La, and Te (5 wt.% Te in-cooperated LCO). Core-level XPS spectra verify the real valence states of the individual components. Co 2p core level spectra, both pure and Te in-cooperated, are shown in Figure 5c,f, respectively. LCO’s two asymmetric peaks are located at 780.18 and 795.45 eV and are most similar to those of Co 2p_3/2_ and Co 2p_1/2_. Paramagnetic Co^2+^ is formed at the surface, accounting for the visible satellite peaks above the primary photo peaks [34]. Because of the presence of numerous excitations and the coexistence of Co^3+^ and Co^2+^ states, the Co 2p peak expanded and shifted toward higher energy when Te was present in conjunction with LCO. The existence of both high-spin Co^2+^ and low-spin Co^3+^ ions was further validated by the appearance of the Co 2p_3/2_ peaks at 781.23 eV with decreased satellite peaks [35,36]. The Co 2p photoelectron spectra of Te-LCO are chemically shifted sufficiently to permit chemical identification. The Co 2p_3/2_ and Co 2p_1/2_ lines for Te-LCO were chemically shifted to 1.05 eV and 1.14 eV, respectively, to higher binding energies than for LCO. The Co 2p_3/2_ has diminished satellite peak may be due to the good exposure of LCO when we mechanically ball mill with Te, which leads to further oxidation of the samples towards the surface. According to research by Frost and colleagues [34,37], photoelectron spectra of high-spin Co^2+^ compounds show robust satellites, while those of low-spin Co^2+^ compounds either show weak satellites or none at all. Te-LCO has a core photoemission peak separation of 15.36 eV, while LCO’s is 15.27 eV. These values are fairly close to the CoO and Co_3_O_4_ values that have previously been published [34,38] as depicted in Figure 5. As anticipated, the La 3d line shows the four component peaks (Figure 5b,e). According to our prior research [19], La 3d_5/2_ and La 3d_3/2_ are associated with the double peaks of LCO, which emerge at 834.7 and 838.34 eV, respectively. These peaks can be attributed to the La^3+^ state and represent charge transfer between the La_2_O_3_, O2p, and La4f orbits [36,37,38,39,40,41] or substantial electronic configuration final state mixing [42]. Te in-corporation causes a shift toward greater binding energies, indicating that the lattice structure is being destroyed. Because of Te’s multi-valence state, the additive peak with the highest intensity at 837.34 eV could be attributable to the synthesis of La sub-oxide (La-O_x_). The observed core-level Te 3d spectra have been deconvoluted into single 3d_3/2_ and 3d_5/2_ spin-state peaks at about 587 and 576 eV, respectively, to help explain it better. The chemical shift seen at Te 3d_5/2_ of approximately 3 eV towards higher binding energy from Te^2−^ is characteristic of NaBH_4_-reduced TeO_2_, which may form some surface oxidation layers (Te^0^). Figure 5i shows the two asymmetric peaks of high-resolution Te 3d XPS spectroscopy, which reveals the Te^4+^/Te^2−^/Te^0^ oxidation state for the NaBH_4_-reduced TeO_2_. The binding energy difference between 3d_3/2_ and 3d_5/2_ is 10.4 eV, which agrees with the literature [43,44,45]. As was previously mentioned, the presence of Te at high oxidation states reconstructs the crystal structure and produces more Co^2+^ and Co^3+^ at various spin states, which leads to additional defects. The perovskite’s A and B sites are both affected by the equally distributed Te^4+^, which results in an increase in the amount of La^2+^ in the A site and anisotropic Co^2+^ in the B site to balance the charge. As a result, the magnetic state of the perovskite LCO was affected by the discrete La 3d and Co 2p photoemission peaks that the 5% Te in-corporated LCO displayed at various spin states [46,47]. The same O 1s peaks associated with the LCO and 5% Te-LCO provide evidence that Te played a part in the incorrect perovskite oxide formation. As demonstrated in Figure 5, both the pristine and Te-LCO O 1s peaks had three unique peaks (h and g). O 1s peaks were observed at 528.8, 530.9, and 532.6 eV on the LCO surface. This major signal at 528.8 eV can be attributable to either bulk oxide or lattice oxygen (O^2−^), which is in line with the findings from earlier studies [41]. When looking at the oxide system, the broad peaks that have greater binding energies are the ones that are the most challenging to interpret. The value of 530.9 eV can be explained by the presence of chemisorbed oxygen (O^−^) or adsorbed H_2_O/OH^−^ species, both of which create a vacancy at the surface for oxygen to occupy [48]. O 1s XPS spectra of the Te-impregnated LCO surface show that superoxide (O^2−^) formation occurs at a peak energy of 532.6 eV, while the other two peaks can be attributed to lattice oxygen, chemically adsorbed oxygen species on the oxygen vacancies, and physically adsorbed oxygen species on the surface. Three peaks with centres at BE = 529.5, 531.6, and 532.7 eV were revealed by deconvolution of the Te-impregnated LCO asymmetric O 1s spectra (Figure 5g) [49]. These peaks correspond to lattice oxygen (such as O_2_), chemically adsorbed oxygen (such as O^−^), and physically adsorbed oxygen (H_2_O and O_2_), respectively. This encourages the conception of more peroxide/superoxide ions, which results in the acquisition of stronger peaks at 532.7 eV. Te^4+^ is in higher oxidation states in the Te in-corporated LCO as a result of the exchange of oxygen species that occurs as oxygen vacancies form between bulk and surface oxides [40].

### 2.5. Magnetic Properties of Te-Incorporated LaCoO_3_

At 300 K, the field dependency of magnetization (M-H) for pristine LCO, 2.5% Te-LCO, and 5% Te-LCO was measured for a field change from −2.5 T to 2.5 T. The M-H loop at 300 K (Figure 6a) suggests that the samples are originally in a paramagnetic state (LCO) and that the magnetic state changes to a weak ferromagnetic with the addition of Te, as shown in Figure 6a, where the hysteresis develops with Te impregnation of LCO. When compared to the parent LCO, the overall magnetization value of the incubated samples is lower. The samples have a very low magnetic saturation (M_S_) value after incorporation with Te. However, some previous studies relate the magnetic state to the anti-ferromagnetic exchange interaction in the samples [50], which arises from the antiferromagnetically ordered localized high spin states present in these systems with the incorporation of Te [51]. The values of magnetization (M_S_), M_R_ and H_C_ extracted from the magnetic data (Figure 6), are given in Table 2. With the initial incorporation of 2.5% Te, it induces coercivity and it decreases for higher concentrations of Te (5%). The anti-symmetric exchange interaction arises from the interaction of low spin Co (III) with the excited high spin Co (III). This further influences magnetocrystallite anisotropy and reduces the coercive field [52,53,54,55]. In our previous study of Mn-doped LCO, the RT M-H shows only a paramagnetic state even after incorporation with Mn at the Co site [19]. From the XPS study, it can be understood that the different valence states of Te may occupy La/Co sites, which enhances the magnetic property in the present LCO system. 

### 2.6. Transmission Analysis of Te-Incorporated LaCoO_3_

Furthermore, TEM was used to verify the nanoparticle size and material structure. LCO Figure 7a,b, 2.5 wt.% Te-LCO (Figure 7d,e) and 5 wt.% Te-LCO (Figure 7g,h) demonstrate an evident ellipsoid structure with a 100–200 nm distribution in Figure 7. Due to the low Te concentration in LCO, lattice plane separation cannot be observed. In addition, Figure 7 shows the 5% Te-LCO’s selected area electron diffraction (SAED) patterns. The composite reflected a single crystal structure with a few irregular brilliant spots, showing that the addition of Te affected the original structure of LCO [56]. Figure 7 depicts the interplanar spacing (d_hkl_) measured from the SAED patterns.

## 3. Materials and Methods

### 3.1. Materials

#### 3.1.1. Materials for the Preparation of Te (Method 1)

The precursor materials used for the preparation of Te are Tellurium dioxide (TeO_2_) (Sigma-Aldrich, Germany), and sodium borohydride (NaBH_4_) (Merck, USA).

#### 3.1.2. Materials for the Preparation of LaCoO_3_ (Method 2)

The precursor materials used for the preparation of LCO are Lanthanum (III) nitrate hexahydrate (La(NO_3_)_3_ 6H_2_O) (Merck), cobalt (II) nitrate hexahydrate (Co(NO_3_)_2_ 6H_2_O) (Merck), ammonia solution about 25% (NH_4_OH) (Merck), and sodium hydroxide (NaOH) (Merck). 

#### 3.1.3. Materials for the Preparation of Te (2.5%)-Impregnated LaCoO_3_ and Te-(5%) Impregnated LaCoO_3_ (Method 3)

The materials obtained after the preparation of method 1 and method 2 are Te and LaCoO_3_. These two materials are used to prepare Te-impregnated LaCoO_3_. All of the chemicals are reagent grade with high purity (99.9%) and can be used straight out of the package.

### 3.2. Methods

#### 3.2.1. Method 1: The Preparation of Te

From room temperature (RT) to boiling point (BP), 100 mL of C_2_H_6_O_2_ and TeO_2_ (2 g wt.) were homogenously mixed using a magnetic stirrer in a silicon oil bath to maintain a temperature of 180 °C in a double-neck round-bottomed flask equipped with a reflux cooler for 2 h or longer. The ratio of salt to reductant was maintained at 1:4. After 2 h, 8 g of NaBH_4_ was added slowly into the mixture with the evolution of gases. While introducing NaBH_4_, regular personnel protective equipment (PPE) precautions and other safety measures were taken. After adding the reducing agent, the reaction mixtures were stirred for two hours and cooled to RT naturally. The ultimate precipitate was centrifuged and repeatedly washed with DI water, acetone, and ethanol before being dried at RT Equation, which represents the reaction of the sodium borohydride and tellurium dioxide (1).
(1)$${TeO}_{2}+C_{2}H_{6}O_{2}\overset{180\;{}^{\circ}C||{NaBH}_{4}}{\to }TeO(OH)+\frac{1}{2}H_{2}^{↑}+\frac{1}{2}B_{2}H_{4}+NaOH$$

TeO (OH^−^) is formed in two steps Equation (1) involves reduction by hydrogenation. Initially, TeO_2_ was reduced to equivalent ions and hydrogenated oxygen (H^+^O) was collected at the surface. The evolved H_2_ may dissociate into H atoms or ions during the reduction process.

#### 3.2.2. Method 2: The Preparation of LaCoO_3_

We have briefly discussed the hydrothermal synthesis of LCO in our previous report [18].

#### 3.2.3. Method 3: The Preparation of Te-Impregnated LaCoO_3_

The third method is ball milling of the samples obtained from the first two methods. The ball-milling of Te and LCO is as follows: 2.5% and 5% Te are mixed with 97.5% and 95% LCO by weight, respectively, and well ground for 10 h with intervals of 30 min to avoid settling of powder on the sides of the ball-mill container. After the ball milling, the material is taken for further studies.

### 3.3. Characterisation Technique

To investigate the phase transformations at ambient temperatures, CuK-alpha radiation was utilised in an Empyrean Malvern Panalytical X-ray Powder Diffractometer. A field emission scanning electron microscope (FESEM) equipped with energy-dispersive X-ray spectroscopy (EDAX) was used to investigate the surface topography and composition of the microstructure. The vibrational modes of the materials are studied using a confocal micro-Raman microscope (WiTec Alpha 300, Germany) equipped with AFM imaging and a He-Ne laser as the excitation source (λ-exc = 532 nm) in a backscattering setup. In order to investigate the chemical composition of the surface, we used X-ray photoelectron spectroscopy (XPS) with Al K-alpha X-rays (Thermo Scientific, UK). A JEOL JEM 2100 HRTEM was utilised for microscopic examination and evaluation. The effect of a magnetic field strength of 5 T on the material’s magnetization (M-H) at room temperature was investigated using SQUID-MPMS (Quantum Design, USA).

## 4. Conclusions

The findings provide a systematic examination of Te impregnation in LCO perovskite oxide at various concentrations. As-prepared perovskite oxides have been analysed for their magnetic properties and crystal structures. The results showed that the samples exhibited an imperfect rhombohedral crystal structure. Adding Te to LCO produces a perovskite with a high multi-valence state of Te^4+/2−^ and three distinct oxygen species, which creates oxygen deficiency. Raman spectroscopy and XPS analysis both show that the presence of Te^4+^ prevents the formation of defective cobalt oxide. After being subjected to impact and iteration in a ball mill, the SEM micrograph of LCO and Te-impregnated LCO demonstrates the structural similarity between the parent and included samples. However, XRD analysis revealed that Te-LCO was successfully reduced and incorporated into the mixture as a result of the ball mill’s influence on the crystallite structure. The structural integrity of the perovskite samples has also been verified using transmission electron microscopy investigation. The presence of weak ferromagnetic order at ambient temperature after impregnation implies its potential application in magnetic devices and hypersonic vehicles.

## Figures and Tables

**Figure 1 ijms-24-10107-f001:**
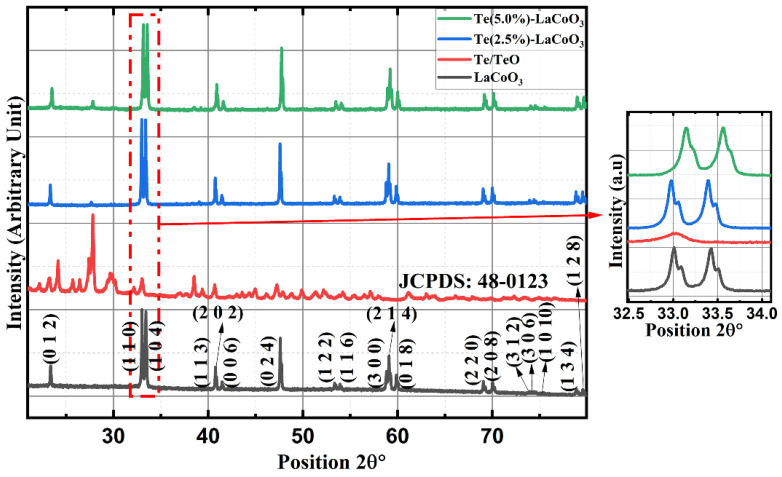
X-ray diffraction of LCO, Te, and Te (2.5% and 5%)-LCO. Outset shows the magnified image of high-intensity peaks of LCO, Te, and Te (2.5% and 5%)-LCO.

**Figure 2 ijms-24-10107-f002:**
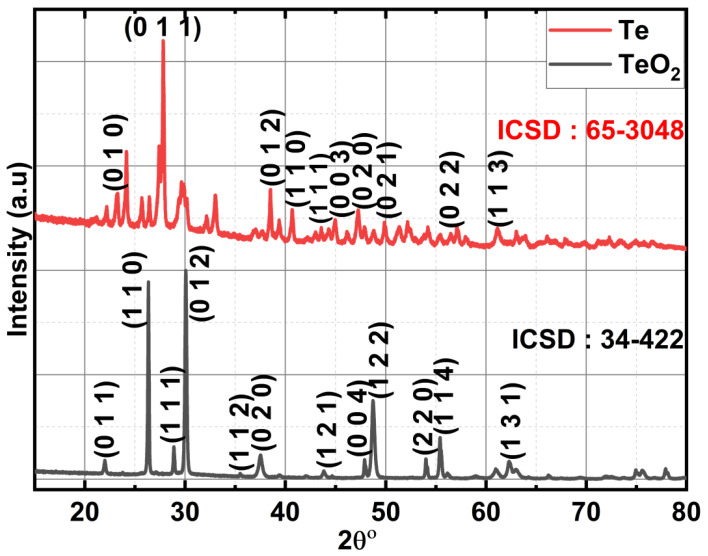
X-ray diffraction of TeO_2_ and Te.

**Figure 3 ijms-24-10107-f003:**
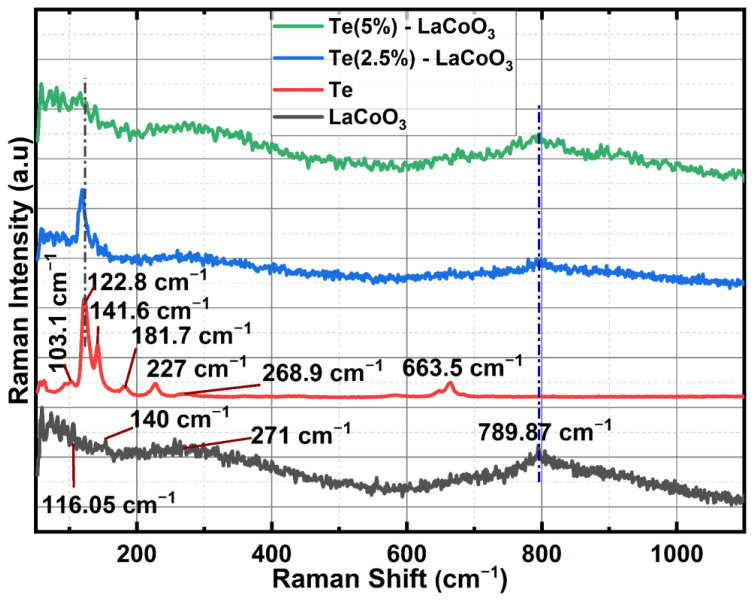
Raman spectroscopy of LCO, Te, 2.5% Te-LCO, and 5% Te-LCO.

**Figure 4 ijms-24-10107-f004:**
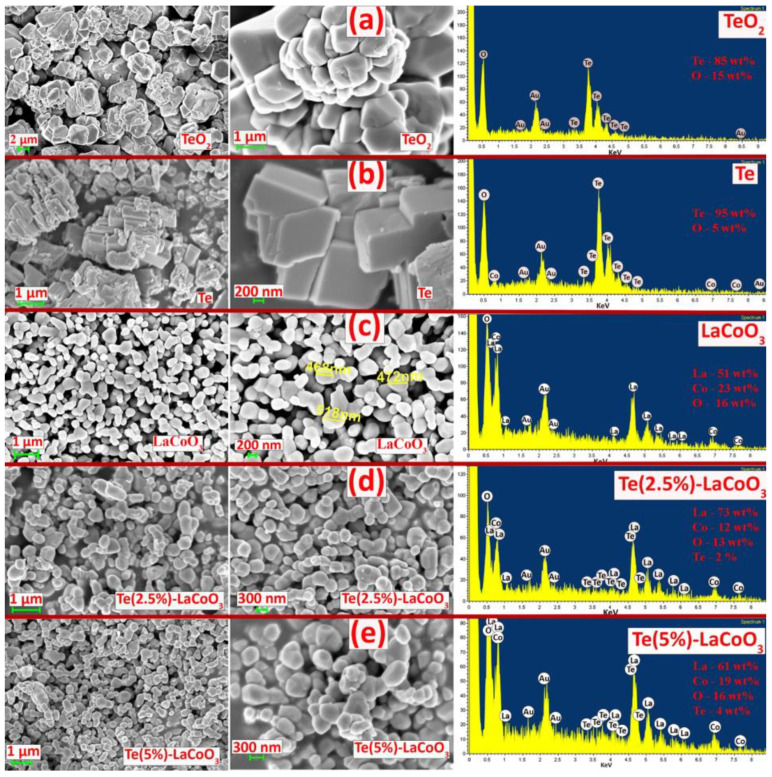
SEM and EDAX image of (**a**) TeO_2_, (**b**) Te, (**c**) LaCoO_3_, (**d**) Te (2.5%)-LCO and (**e**) Te (5%)-LCO.

**Figure 5 ijms-24-10107-f005:**
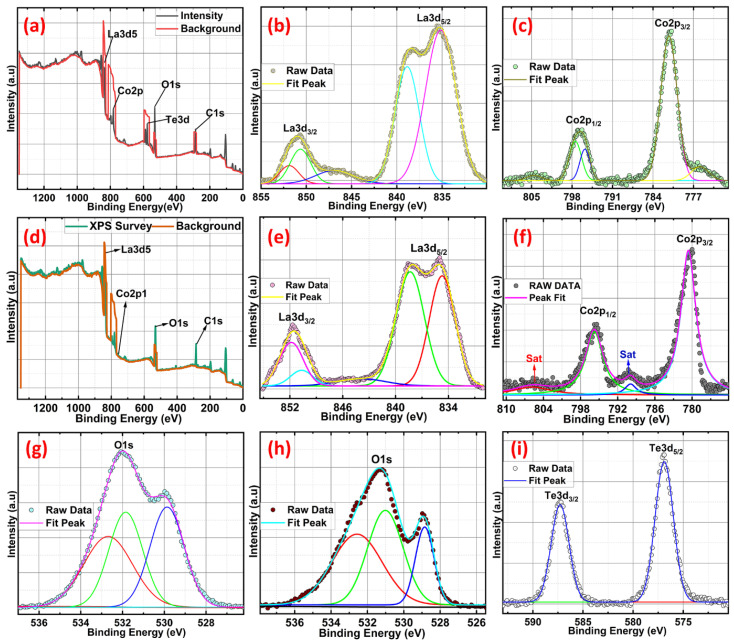
XPS survey scan (**a**,**d**) of LCO and Te (2.5%)-LCO, respectively. Detail XPS scan of La 3d (**b**,**e**), Co 2p (**c**,**f**), O 1s (**g**,**h**) of LCO and Te (2.5%)-LCO, respectively, and Te 3d (**i**) of Te (2.5%)-LCO.

**Figure 6 ijms-24-10107-f006:**
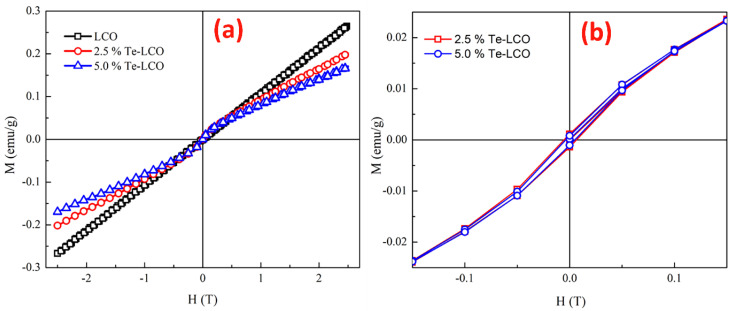
Isothermal magnetization curve of (**a**) LCO, 2.5% Te-LCO, and 5% Te-LCO at RT, and (**b**) enlarged image of LCO, 2.5% Te-LCO, and 5% Te-LCO at RT.

**Figure 7 ijms-24-10107-f007:**
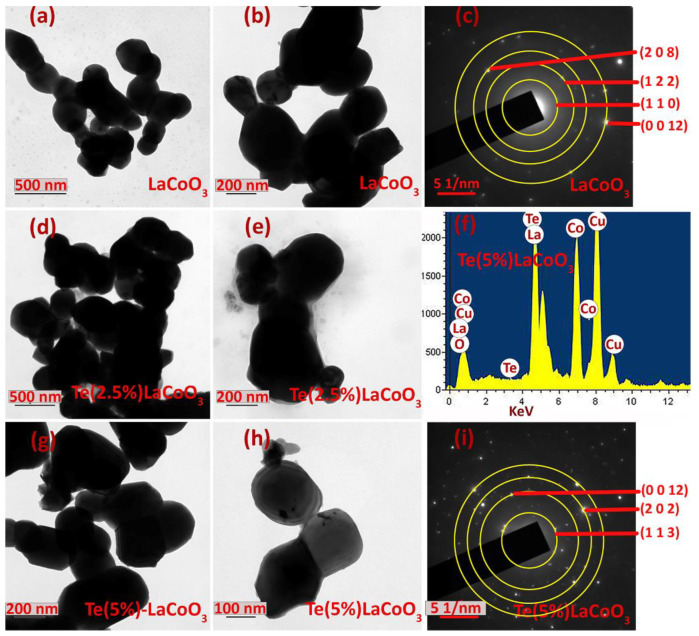
TEM image of (**a**,**b**) LaCoO3; (**d**,**e**) Te (2.5%) LCO; and (**g**,**h**) Te (5%) LCO. SAED of (**c**) LaCoO_3_ and (**i**) Te (5%) LCO and EDS image of (**f**) Te (5%) LCO.

**Table 1 ijms-24-10107-t001:** The crystallite size and lattice parameter of TeO_2_, Te, LCO, LCO/Te (2.5%), and LCO/Te (5%).

Composition	Crystal System	Space Group	**Crystallite Size (nm)** $D=\frac{0.9\lambda }{\beta cos\theta }$	Lattice Parameter (Å)		
				a	b	c
TeO_2_	Tetragonal	P4_1_2_1_2	26.80514	5.40681	5.40865	13.20149
Te	Hexagonal	P3_1_21	28.33724	4.19097	4.19123	5.98099
LaCoO_3_	Rhombohedral	$R\overset{-}{3}$c	43.42469	5.41768	5.39366	13.13084
2.5% Te/LCO	Rhombohedral	$R\overset{-}{3}$c	31.59925	5.42428	5.39256	13.1579
5% Te/LCO	Rhombohedral	$R\overset{-}{3}$c	31.59925	5.42681	5.39434	13.20149

**Table 2 ijms-24-10107-t002:** Saturation magnetization (M), remnant magnetization (M_r_), and coercivity (H_C_) of pure and Te@LCO.

Composition	H_C_ (T)	M (emu/g) @300 K	M_r_ (emu/g)
LCO	--	---	-----
2.5% Te-LCO	0.0065	0.20166	0.00128
5% Te-LCO	0.0049	0.17051	0.00094

## Data Availability

All the data used in the manuscript are within the manuscript.

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
