# Peer review of "Influence of Te-Incorporated LaCoO3 on Structural, Morphology and Magnetic Properties for Multifunctional Device Applications"

_ijms, 2023, doi:10.3390/ijms241210107_

Round 1
Reviewer 1 Report
1. According to the abstract, the preparation and characterizations, however, should you write one sentence regarding the objective of the statement here? The research gap is unclear. You should highlight the research gap and what you are doing to close it.
2. After the reference [25], you can highlight the impact of the incorporated LaCoO3 for a few lines in the introduction and how it differs from the previous study.
3. Subdivide the materials and methods into First Materials (what materials are used and purchased) and Preparation/Characterization.
4. The outcome and discussion should be further subdivided or subtitled according to certain characterization standards. The characterization results are clear; however, they must be thoroughly discussed and summarized.
5. When compared to the parent LCO, Te-LCO is low; However, what does advantage of Te incorporation explain.
6. In conclusion, the sentence state that “Te to LCO, a perovskite is created that has a high valence state of Te4+/2-, However in the abstract “Te4+/2+. check throughout manuscript.
7. The format of all of the references should be unified. Some journals are abbreviated, while others are not.) and I recommend that you look for some recent articles about this work.
8. In table 1, I have seen the format errors “TeO2’’, Lattice parameter “a, b, c” (Table-1)
9. In figures, Figure 7 should be redrawn because the drawing style is inconsistent.
10. I have seen many format errors in the whole manuscript, room temperature (r.t.) may be (RT); Raman spectra. The weedy bands, O2p and La, O1s peaks and etc.,
1. 1. Please check the grammar and typing errors of the manuscript.
Author Response
Response to the reviewers’ comments
Title: Influence of Te incorporated LaCoO3 on structural, morphology and magnetic properties.
Journal: Journal of Molecular Sciences
Manuscript ID: ijms-2442728
We thank reviewer for evaluating the manuscript and providing valuable inputs for this manuscript. The detailed response to the pointed concern is as:
Reviewer #1:
Question: According to the abstract, the preparation and characterizations, however, should you write one sentence regarding the objective of the statement here? The research gap is unclear. You should highlight the research gap and what you are doing to close it.
Reply: The paramagnetic nature of LCO at RT persist even after doping Mn in B-site. So we choose LCO to improve the RT magnetic properties as it is cheap materials for advanced multi-functional and energy applications.
Question: After the reference [25], you can highlight the impact of the incorporated LaCoO3 for a few lines in the introduction and how it differs from the previous study.
Reply: We have included the impact of the incorporation of LCO and how it differs from our previous study introduction part after reference [25].
Question: Subdivide the materials and methods into First Materials (what materials are used and purchased) and Preparation/Characterization.
Reply: We have subdivided the materials and methods into materials used, preparation method and characterization.
Question: The outcome and discussion should be further subdivided or subtitled according to certain characterization standards. The characterization results are clear; however, they must be thoroughly discussed and summarized.
Reply: We have changed the results and discussion by subdividing the characterization standard and thoroughly discussed and summarized.
Question: When compared to the parent LCO, Te-LCO is low; However, what does advantage of Te incorporation explain.
Reply: When compared to parent LCO the Te-LCO there is a transformation of paramagnetic LCO weak ferromagnetic nature.
Question: In conclusion, the sentence state that “Te to LCO, a perovskite is created that has a high valence state of Te4+/2-, However in the abstract “Te4+/2+. check throughout manuscript.
Reply: The correction has been made in the manuscript.
Question: The format of all of the references should be unified. Some journals are abbreviated, while others are not.) and I recommend that you look for some recent articles about this work.
Reply: Reference has been corrected as suggested.
Question: In table 1, I have seen the format errors “TeO2’’, Lattice parameter “a, b, c” (Table-1)
Reply: The format error has been rectified in the manuscript.
Question: In figures, Figure 7 should be redrawn because the drawing style is inconsistent.
Reply: Redrawn diagram has been updated in the figure.
Question: I have seen many format errors in the whole manuscript, room temperature (r.t.) may be (RT); Raman spectra. The weedy bands, O2p and La, O1s peaks and etc.,
Reply: The format errors were rectified and updated in the manuscript.
We revised the paper as per reviewer suggestion and incorporation in the revised manuscript mention in the blue color. We believe that after introducing the inputs/suggestions by the reviewer, the manuscript has improved a lot and is now suitable for publication.

Reviewer 2 Report
The article studies the influence of Te incorporated in LaCoO3 on the structural, morphological and magnetic properties.
The paper presents the results on the synthesis of the material Te-LaCoO3, the study of the structural, morphological and magnetic properties. All obtained results are presented in detail.
The paper can be published in present form without changes.
There are no technical remarks on the design of the article. It may only be advisable to reduce the fonts along the axes of the drawings and captions in the drawings.
The main purpose of this article is to investigate the effect of Te incorporated in LaCoO3 on the structural and ferromagnetic properties of perovskite. Perovskites can have a wide range of properties (piezoelectrics, paramagnetics, magnetics). The article demonstrates that incorporation of Te allows ferromagnetic properties to appear.
The work is original and deserves research attention, as it opens up new prospects for the application of modified рerovskites. In general, the paper proposes the possibility of modifying perovskites by incorporating different elements. In the future, there are opportunities to create new compositions with unique physical properties.
The article presents methods for the synthesis of the material Te: LaCoO3 in some detail. An accessible method for obtaining perovskite with ferromagnetic properties is found.
As for the study of the material, in principle, everything is done very correctly. But in the future it would be interesting to investigate the structural factors in the material by high resolution X-ray diffraction, since the crystallographic structure is polar. It would also be interesting to investigate magnetic dichroism in the material under external magnetic fields. But this is more of a wish in the aspect of continuing this research.
The conclusions in the article are consistent with its content. The references are correct.
Тables and figures correspond to the content of the article. A comment should be made here regarding the design of the figures. It may only be advisable to reduce the fonts along the axes of the drawings and captions in the drawings.
OK
Author Response
Response to the reviewers’ comments
Title: Influence of Te incorporated LaCoO3 on structural, morphology and magnetic properties.
Journal: Journal of Molecular Sciences
Manuscript ID: ijms-2442728
We thank reviewer for evaluating the manuscript and providing valuable inputs for this manuscript. The detailed response to the pointed concern is as:
Reviewer #2:
The article studies the influence of Te incorporated in LaCoO3 on the structural, morphological and magnetic properties. The paper presents the results on the synthesis of the material Te-LaCoO3, the study of the structural, morphological and magnetic properties. All obtained results are presented in detail. The paper can be published in present form without changes.
There are no technical remarks on the design of the article. It may only be advisable to reduce the fonts along the axes of the drawings and captions in the drawings. The main purpose of this article is to investigate the effect of Te incorporated in LaCoO3 on the structural and ferromagnetic properties of perovskite. Perovskites can have a wide range of properties (piezoelectrics, paramagnetics, magnetics). The article demonstrates that incorporation of Te allows ferromagnetic properties to appear.
The work is original and deserves research attention, as it opens up new prospects for the application of modified рerovskites. In general, the paper proposes the possibility of modifying perovskites by incorporating different elements. In the future, there are opportunities to create new compositions with unique physical properties.
The article presents methods for the synthesis of the material Te: LaCoO3 in some detail. An accessible method for obtaining perovskite with ferromagnetic properties is found.
As for the study of the material, in principle, everything is done very correctly. But in the future it would be interesting to investigate the structural factors in the material by high resolution X-ray diffraction, since the crystallographic structure is polar. It would also be interesting to investigate magnetic dichroism in the material under external magnetic fields. But this is more of a wish in the aspect of continuing this research.
The conclusions in the article are consistent with its content. The references are correct.
Question: Тables and figures correspond to the content of the article. A comment should be made here regarding the design of the figures. It may only be advisable to reduce the fonts along the axes of the drawings and captions in the drawings.
Reply: We thank the reviewer to review our manuscript and giving valuable comments. As per the reviewer suggestions, the image font size modified and incorporated in the revised manuscript.
We believe that after introducing the inputs/suggestions by the reviewer, the manuscript has improved a lot and is now suitable for publication.
